# Effects of Vibrant Soundbridge on tinnitus accompanied by sensorineural hearing loss

**Jeon Mi Lee**[1], **Hyun Jin Lee**[2], **In Seok Moon**[3]*, **Jae Young Choi**[3]

**1** Department of Otorhinolaryngology, Ilsan Paik Hospital, Inje University College of Medicine, Goyang, Korea, **2** Department of Otorhinolaryngology, Incheon St. Mary's Hospital, The Catholic University of Korea, Bupyeong, Incheon, Korea, **3** Department of Otorhinolaryngology, Yonsei University College of Medicine, Seoul, Korea

* ismoonmd@yuhs.ac

**Data Availability Statement:** All relevant data are within the manuscript and its Supporting Information files.

**Funding:** This study was supported by a grant from the National Research Foundation of Korea

## Abstract

### Objectives

Tinnitus is a common symptom among patients with hearing loss, and many studies have reported successful tinnitus suppression with hearing devices. Active middle ear implantation of the Vibrant Soundbridge (VSB) is a good alternative to existing hearing devices. This study evaluated the effects of VSB implantation on tinnitus and sought to identify the main audiological factor that affects tinnitus suppression.

### Methods

The study participants were 16 adults who had tinnitus with sensorineural hearing loss, and who underwent VSB implantations. Pure-tone audiometry; word recognition test; tinnitus handicap inventory (THI); and visual analog scale (VAS) assessment of loudness, awareness, and annoyance were performed before and 12 months after surgery. Changes in hearing threshold, word recognition scores (WRS), THI scores, and VAS scores were analyzed.

### Results

VAS scores for loudness (mean difference: 1.9, 95% CI: 0.6, 3.1), awareness (mean difference: 1.6, 95% CI: 0.4, 2.8), and annoyance (mean difference: 1.7, 95% CI: 0.7, 2.8) showed significant improvements from baseline to 12 months after surgery. In addition, THI scores showed a significant decrease (mean difference: 13.8, 95% CI: 2.9, 24.9). The average hearing threshold level, WRS, and most comfortable level (MCL) also showed significant improvements at 12 months after surgery (mean difference: 17.3, 95% CI: 13.3, 21.3; mean difference: −7.6, 95% CI: −15.1, −0.1; mean difference: 26.3, 95% CI: 22.9, 29.6, respectively). Among the aforementioned factors, changes in MCL were best correlated with those in THI scores (mean difference: 2.55, 95% CI: 0.90, 4.21).

### Conclusion

A VSB implant is beneficial to subjects with tinnitus accompanied by sensorineural hearing loss. The changes in THI scores best correlated with those in MCL. This improvement may

(NRF), funded by the Ministry of Education
(2018R1D1A1A02085472) to I.S.M.

**Competing interests:** The authors have declared
that no competing interests exist.

represent a masking effect that contributes to tinnitus suppression in patients with VSB
implants.

## Introduction

Tinnitus is the perception of sound in the absence of sound stimuli, and affects approximately
10–15% of the adult population [1], with 20% of those affected experiencing a significant
decrease in quality of life [2]. Traditionally, tinnitus was considered an otological disorder, but
recent advances in neuroimaging have shifted the perspective toward the neural correlates
underlying different forms of tinnitus [3, 4]. Increased pathophysiological understanding of
tinnitus has contributed to various treatment attempts, including pharmacological treatments,
auditory stimulation, psychological treatments, and brain stimulations.

Many studies have reported tinnitus suppression following auditory stimulation with hear-
ing devices, such as hearing aids (HAs), cochlear implants, and more recently, middle ear
implants. As early as 1947, Saltzman and Ersner reported that patients with tinnitus benefited
from the use of HAs [5], and the efficacy of this approach has been confirmed by other studies.
Surr et al. reported that approximately 50% of patients with HAs experienced relief from tinni-
tus [6], and Folmer and Caroll reported the improvement of tinnitus with HAs in 70% of par-
ticipants [7]. In 2008, Trotter et al. reported a 25-year study examining the effects of HAs on
tinnitus [8], in which 350 of 826 patients with HAs (42.4%) reportedly showed effective sup-
pression of tinnitus. Since Van de Heyning, in 2008 [9], reported that patients with tinnitus
resulting from single-sided deafness benefited from the use of cochlear implants, many studies
have supported the beneficial effects of cochlear implants on tinnitus accompanied by sensori-
neural hearing loss. Amoodi and colleagues reported a suppressive effect on tinnitus in 66% of
implant users in their retrospective study [10], and Bovo et al. reported loudness and annoy-
ance suppression in 36.1% and 30.6%, respectively, of 36 patients with cochlear implants in
their prospective study [11]. Other prospective studies have reported a reduction in tinnitus
intensity in 92% of implant users [12], and a reduction of tinnitus handicap inventory (THI)
scores in 65% of implant users [13]. Thus far, only one paper has reported the effects of the
VSB on tinnitus [14]; in that study, VAS assessments of patients showed 55%, 45%, and 55%
improvements in loudness, awareness, and annoyance, respectively after VSB implantation.
Although these studies used different methods to assess tinnitus reduction, it is apparent that
use of hearing devices have some therapeutic effects on tinnitus.

Several possible mechanisms can explain tinnitus suppression by means of hearing devices.
One possible mechanism involves hearing gain. Recent neuroimaging studies have revealed
that tinnitus brain networks include not only sensory auditory areas, but also cortical regions
involved in perceptual, emotional, memory, attentional, and salient functions [15, 16]. Emo-
tional stress caused by hearing loss also affects the severity of tinnitus, and hearing gain by
means of hearing devices can help overcome this emotional stress. Another proposed mecha-
nism is the creation of a masking effect. The use of hearing devices reduces patients' awareness
of their tinnitus by directing their attention to external auditory stimuli [7, 8]. As the level of
external sounds increases with hearing devices, the patients' perception of tinnitus would
decrease. Lastly, secondary plastic reorganization in the central auditory system reduces tinni-
tus. Auditory input is decreased in subjects with hearing impairment, and consequently neural
firing rate and neural synchrony increases, resulting in the plastic reorganization of the audi-
tory cortex, with subsequently sustained awareness of tinnitus [3, 15]. Increasing the external

auditory input with hearing devices can induce secondary plasticity and decrease the patients' perception of tinnitus [17–19].

The Vibrant Soundbridge (VSB, Med-El, Innsbruck, Austria) is a partially implanted, active middle ear device. It directly delivers mechanical vibrations to the middle ear by placing the floating mass transducer (FMT) on the middle ear structures. Previous studies have confirmed that it confers superior, more usable amplification, as well as easier communication in daily listening environments than conventional amplification via HAs [20]. Accordingly, we hypothesized that the use of VSB as a hearing device would have a positive association on tinnitus in patients with sensorineural hearing loss.

The purpose of the present study was to investigate the effects of VSB on tinnitus and to identify the main audiological factor involved in the changes in tinnitus, with a view to improving the treatment of tinnitus in patients with sensorineural hearing loss.

## Materials and methods

### Subjects

This study involved a retrospective chart review of patients who had sensorineural hearing loss with tinnitus, and who underwent VSB implantation at the Severance Hospital in Seoul, Korea, during January 2016 to July 2019. Inclusion criteria were 1) sensorineural hearing loss accompanied by ipsilateral tinnitus, 2) VSB implantations performed in the affected ears, 3) stable chronic tinnitus that did not respond to any previous treatments, and 4) follow-up for more than 1 year after VSB implantation and completion of audiological tests and self-assessment questionnaires preoperatively and at 1 year postoperatively. Exclusion criteria were 1) chronically disabled middle ears or previous ear surgeries, 2) fluctuating hearing and/or tinnitus, and 3) known conditions that could cause hearing loss and/or tinnitus (e.g., Meniere's disease or vestibular schwannomas).

In total, 47 patients underwent VSB implantation between January 2016 and July 2019. From these, 3 patients were excluded because they had mixed hearing loss. In addition, 16 patients who did not experience tinnitus, 4 who were followed up for less than 1 year, 7 who did not complete the questionnaires, and 1 foreigner for whom a speech discrimination test was not available were excluded. Finally, 16 subjects (7 males, 9 females) were enrolled in this study. The mean age at operation of the 16 patients was 66.0 ± 8.2 (53–76) years. All patients had moderate to severe sensorineural hearing loss (58.9 ± 6.9 dB HL) accompanied by subjective tinnitus on the same side. Hearing loss was bilaterally symmetric in 12 patients, and asymmetric in 4 patients. Those 4 patients had not recovered from previous sudden unilateral hearing loss, and the implantations were performed in the worse ears. The configuration of hearing loss in the implanted ears was flat in most patients (13, 81.3%), followed by sloping (2, 12.5%) and ascending (1, 6.2%). The patients' demographic data are listed in Table 1.

All procedures were performed in accordance with the 1975 Declaration of Helsinki. The study was approved by the institutional review board of the Severance Hospital in Seoul, Korea (4-2017-0847). All of the subjects provided informed consent to participate in the study.

### Audiological assessment

All patients were assessed by using pure-tone audiometry and word recognition tests (word recognition score; WRS) at preoperative and 12-months postoperative time points. The pure-tone air (250–8000 Hz) and bone conduction (250–4000 Hz) thresholds were measured using clinical audiometers in a double-walled audio booth. The pure-tone average (PTA) was defined as the mean value of the measurements taken at frequencies of 500 Hz, 1 kHz, 2 kHz, and 4 kHz. A word recognition test was performed to obtain the maximal WRS, which was

**Table 1. Subject demographic data (N = 16).**

| Patient no. | Sex/Age | Side of VSB implantation | Hearing-loss symmetry | Hearing-loss configuration | PTA$_4$ of the VSB implanted ear (dB HL) | Tinnitus |
|---|---|---|---|---|---|---|
| 1 | F/73 | Left | Asymmetric | ascending | 72.5 | Left |
| 2 | M/67 | Left | Symmetric | Flat | 60 | Left |
| 3 | F/73 | Right | Symmetric | Flat | 51.3 | Bilateral |
| 4 | F/54 | Right | Symmetric | Flat | 56.3 | Bilateral |
| 5 | F/58 | Left | Symmetric | Flat | 55 | Bilateral |
| 6 | F/76 | Left | Symmetric | Flat | 60 | Left |
| 7 | F/58 | Left | Asymmetric | Flat | 58.8 | Left |
| 8 | M/72 | Right | Symmetric | Flat | 72.5 | Bilateral |
| 9 | M/53 | Right | Symmetric | flat | 53.8 | Bilateral |
| 10 | M/76 | Right | Symmetric | flat | 56.3 | Bilateral |
| 11 | F/54 | Right | Symmetric | flat | 67.5 | Bilateral |
| 12 | M/65 | Left | Symmetric | sloping | 55 | Bilateral |
| 13 | F/70 | Left | Symmetric | flat | 62.5 | Bilateral |
| 14 | M/73 | Left | Asymmetric | flat | 55 | Left |
| 15 | M/69 | Right | Asymmetric | sloping | 48.8 | Bilateral |
| 16 | F/64 | Left | Symmetric | flat | 57.5 | Left |

PTA$_4$ = pure-tone average of four frequency thresholds (0.5, 1, 2, and 4 kHz), VSB = Vibrant Soundbridge

measured at the most comfortable hearing level (MCL) using 50 monosyllabic Korean words that are commonly heard during everyday life. The Korean words were from a validated and standardized resource [21] and were phonetically balanced. The sound was set to a constant intensity through the microphone. The MCL was defined as the sound level at which patients best heard and understood non-emotional sentences in Korean. Measurements of MCL with respect to loudness were obtained using the forced choice method, in which the experimenter manipulated the 5-dB-step attenuator dial until subjects chose the MCL.

Patients were reassessed 12 months after surgery, and the assessments consisted of aided pure-tone audiometry and a word recognition test. Aided pure-tone audiometry was performed after switching on the VSB alone using sound field (air conduction) testing through speakers with contralateral masking noise, which was achieved via a calibrated headphone. The masking procedure was the plateau method, also known as Hood's technique [22].

Functional hearing gain (FHG) was determined as the difference between preoperative and postoperative free-field audiometry data.

## Tinnitus assessment

To assess pre-operative and postoperative tinnitus severity, all patients completed 2 self-administered questionnaires: a visual analog scale (VAS) for loudness, awareness, and annoyance, with ratings from 0 to 10, and the tinnitus handicap inventory (THI) developed by Newman et al. [23]. The questionnaires were conducted at preoperative and 12-months postoperative time points, and involved the same tests and full compliance. During this period, patients were followed up without any tinnitus retaining therapy, and/or any further treatments for tinnitus. We characterized an "improvement" as a decrease of more than 20% in the postoperative scores compared with the preoperative scores, and "worsened" as an increase of more than 20% in postoperative scores compared with the preoperative scores [24].

## Statistical analyses

Statistical analysis was performed with SPSS for Windows, version 21 (SPSS Inc., Chicago, IL, USA). The results of multiple experiments are presented as the mean ± standard deviation. Comparisons were performed between continuous variables using Student's *t*-test or the paired *t*-test for evaluating differences between 2 groups if the data exhibited a normal distribution. Multiple regression analysis was performed to identify which audiological factors affected the improvement of tinnitus using a general linear model. Values of $p < 0.05$ were considered to be statistically significant.

## Results

### Changes in tinnitus

The therapeutic effects of VSB on tinnitus were encouraging (Table 2). The initial average THI score was 55.6 ± 17.2, which decreased to 41.8 ± 24.7 at 12 months after surgery; the decrease was statistically significant ($t = 2.7$, $p < 0.05$). Considering the criterion of a 20% decrease in THI score, 7 patients (43.8%) reported improvement of tinnitus; however, 8 patients (50.0%) reported no change, and 1 patient (6.2%) reported worsening of tinnitus.

The patients were also questioned regarding how tinnitus, in terms of loudness, awareness, and annoyance, affected their lifestyle and emotions, using a symptom-rating scale based on a visual analogue scale (VAS) self-reported questionnaire. The questionnaire results demonstrated statistically significant improvements after VSB implantation. VAS scores for loudness decreased from 6.8 ± 2.3 to 4.9 ± 1.5 ($t = 3.3$, p < 0.01). VAS scores for awareness and annoyance also decreased from 6.3 ± 2.1 to 4.7 ± 1.9 ($t = 2.9$, $p < 0.05$) and from 6.3 ± 2.0 to 4.6 ± 1.8 ($t = 3.4$, $p < 0.01$), respectively. Based on the 20% criterion, 7 patients (43.8%) reported improvement in tinnitus loudness, 8 patients (50.0%) reported no change, and 1 patient

**Table 2. Comparing means of preoperative unaided and postoperative aided ear condition (N = 16).**

| Variable | | | Preoperative unaided ear (A) | Postoperative aided ear (B) | Paired differences | *t*-value | 95% CI of the difference (Lower, Upper) | | p-value |
|---|---|---|---|---|---|---|---|---|---|
| Tinnitus assessment | THI | M | 55.6 | 41.8 | 13.8 | 2.7 | 2.9 | 24.9 | < 0.05 |
| | | SD | (17.2) | (24.7) | (20.6) | | | | |
| | VAS for loudness | M | 6.8 | 4.9 | 1.9 | 3.3 | 0.6 | 3.1 | < 0.01 |
| | | SD | (2.3) | (1.5) | (2.3) | | | | |
| | VAS for awareness | M | 6.3 | 4.7 | 1.6 | 2.9 | 0.4 | 2.8 | < 0.05 |
| | | SD | (2.1) | (1.9) | (2.2) | | | | |
| | VAS for annoyance | M | 6.3 | 4.6 | 1.8 | 3.4 | 0.7 | 2.8 | < 0.01 |
| | | SD | (2.0) | (1.8) | (2.0) | | | | |
| Audiological assessment | PTA (dB HL) | M | 58.9 | 41.6 | 17.3 | 9.2 | 13.3 | 21.3 | < 0.001 |
| | | SD | (6.9) | (6.7) | (7.5) | | | | |
| | WRS (%) | M | 62.1 | 69.8 | -7.6 | -2.2 | -15.1 | -0.1 | < 0.05 |
| | | SD | (10.6) | (13.9) | (14.1) | | | | |
| | MCL (dB HL) | M | 83.5 | 57.3 | 26.3 | 16.6 | 22.9 | 29.6 | < 0.001 |
| | | SD | (5.9) | (3.5) | (6.3) | | | | |

THI = tinnitus handicap inventory, VAS = visual analogue scale, PTA = pure-tone average, WRS = word recognition scores, MCL = most comfortable level, M = mean, SD = standard deviation, CI = confidence interval

(6.2%) reported louder tinnitus after implantation. The proportions were similar in the other 2 questionnaires. For awareness, 8 patients (50.0%) reported improvement, 6 patients (37.5%) found no change, and 2 patients (12.5%) reported worsening. For annoyance, 8 patients (50.0%) reported improvement, while 7 (43.8%) were stable and 1 (6.2%) was more annoyed after implantation than under the unaided condition (see S1 Dataset for all relevant data).

### Audiological assessment

Table 2 shows the changes in audiological factors. Preoperative PTA was 58.9 ± 6.9 dB. After 12 months with the fine-fitting condition, the PTA improved to 41.6 ± 6.7 dB ($t = 9.2$, $p < 0.001$), and the FHG was 17.3 ± 11.9 dB. Word recognition tests were performed under the preoperative unaided condition, and under the VSB-aided condition at 12-months postoperatively. The patients' WRS improved from 62.1 ± 10.6% to 69.8 ± 13.9% ($t = -2.2$, $p < 0.05$,), and their MCL decreased from 83.5 ± 5.9 dB to 57.3 ± 3.5 dB ($t = 16.6$, $p < 0.001$, see S1 Dataset for all relevant data).

### Influencing factors

To identify the audiological factors that improved tinnitus, we performed multiple regression analysis. The 3 audiological factors included were as follows: 1) average FHG, 2) changes in MCL, and 3) changes in WRS. The improvement of tinnitus was measured as 1) changes in THI, 2) changes in VAS for loudness, 3) changes in VAS for awareness, and 4) changes in VAS for annoyance. When we performed multiple regression analysis, only decreased MCL was significantly and positively related to the improvement of THI. Improvement of MCL (unstandardized regression coefficient = 2.55 with standard error = 0.76) explained 60.0% of the THI improvement ($p < 0.01$, Fig 1, Table 3). Changes in VAS scores were also analyzed, but none of the 3 audiological factors showed a significant relationship with changes in VAS scores.

## Discussion

In this study, we demonstrate that the effects of VSBs on tinnitus accompanied by sensorineural hearing loss were beneficial. Within 1 year of VSB fine-fitting, the THI scores and VAS scores for loudness, awareness, and annoyance were significantly decreased. The mechanism underlying the beneficial effect of VSB on tinnitus remains unclear, although it may be similar to that of other hearing devices. In the present study, we also specifically sought to identify the main factor affecting the improvement of tinnitus in our study subjects and found that changes in THI scores were best correlated with the changes in MCLs. As MCL decreased with VSB, THI scores significantly decreased in the subjects.

Lowering MCL is critical in auditory rehabilitation of patients with sensorineural hearing loss, because maximum speech intelligibility is achieved at levels higher than the MCL [25, 26]. Thus, patients with a high MCL have difficulty in understanding routine conversation at moderate sound levels. In the present study, patients showed a significant reduction in MCL, with a significant improvement in WRS. However, the changes in tinnitus were better correlated with the changes in MCL than with changes in WRS. For example, a patient in the present study showed a 34-dB improvement in MCL, but a 12% decrease in WRS. Her THI scores had improved from 52 to 28. Interpretation of these results suggest that the mechanism best explaining the effects of VSB on tinnitus is a masking effect, involving increased external sound perception.

A masking effect is a good rationale for sound therapy, and is a vital component of effective tinnitus management [7]. Its role is to reduce the contrast between the tinnitus signal and background activity in the auditory system, thereby forming a mask that is not limited to

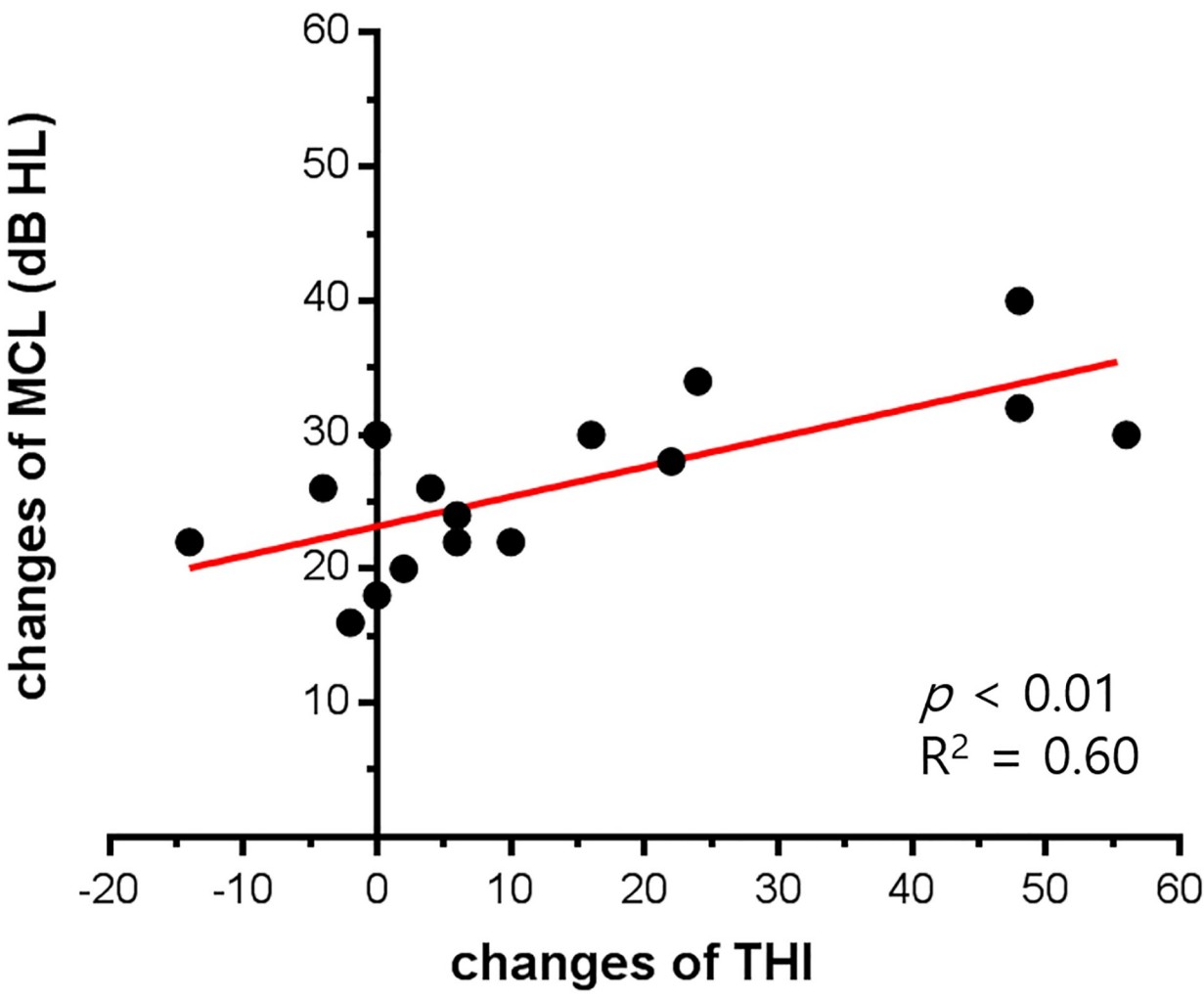

**Fig 1. Correlations between changes in THI and changes in MCL.** Multiple regression analysis revealed that only increased MCL was significantly associated with improvements of THI. Changes in THI was well correlated with changes in MCL ($p < 0.01$, $R^2 = 0.60$).

human conversation. Background music, environmental sounds, or even meaningless white noise can be used as a mask, but Accuracy of speech reception is not the most important criterion for an effective mask, which explains the better correlation of tinnitus suppression with changes in MCL than with changes in WRS.

**Table 3. Multiple regression analysis for the association between audiological improvement and tinnitus improvement (N = 16).**

| Variables | ΔTHI | ΔVAS for loudness | ΔVAS for awareness | ΔVAS for annoyance |
|---|---|---|---|---|
| ΔPTA | -0.41 (-1.88, 1.05) | -0.09 (-0.33, 0.15) | -0.11 (-0.33, 0.12) | -0.13 (-0.33, 0.08) |
| ΔWRS | 0.28 (-0.37, 0.94) | 0.03 (-0.14, 0.08) | 0.02 (-0.13, 0.08) | 0.02 (-0.12, 0.07) |
| ΔMCL | 2.55** (0.90, 4.21) | 0.16 (-0.12, 0.43) | 0.19 (-0.06, 0.45) | 0.17 (-0.07, 0.40) |
| R-squared | 0.60 | 0.12 | 0.18 | 0.18 |

THI = tinnitus handicap inventory, VAS = visual analogue scale, PTA = pure-tone average, WRS = word recognition scores, MCL = most comfortable level, M = mean, SD = standard deviation

Unstandardized regression coefficients are shown, and 95% confidence intervals are given in parentheses. 2-tailed test;

$^{**}p < 0.01$

Assuming that reducing the MCL is an important factor for suppressing tinnitus, the VSB might have an advantage over other HAs used for tinnitus suppression. A recent report compared the MCL in HA-aided and VSB-aided conditions [26] and found that hearing gains were similar under both aided conditions, but that the VSB-aided condition showed better improvement in MCL, to a value comparable to that of the normal hearing population. This phenomenon is explained by a mass effect. A loaded FMT provides system stiffness, and it results in increased hearing gain at mid to high frequencies, and decreased hearing gain at low frequencies [27]. Lowering the MCL is attributable to robust amplifications of the mid to high frequencies [26]; thus, VSB is superior to conventional Has in terms of lowering the MCL. Further studies on tinnitus suppression, comparing HA-aided and VSB-aided conditions could verify the importance of lowering MCL in tinnitus management.

However, in this study, the correlation was found only when the factors were analyzed with THI scores, but not with VAS scores. Several factors can explain these discrepancies. First, the differences in the applied scales could affect the results. The VAS is composed of simple 11-point numeric scales, whereas the THI is composed of 25 questions, which can be answered in 3 ways: Yes, Sometimes, and No. Second, while the VAS only estimates the severity of tinnitus perception, THI assesses its psychological aspects, which are particularly important in tinnitus treatment. Finally, a sample size compromises the power for the detection of treatment effects and increases the possibility of false-negative results. Thus, the small sample size in this study may be a reason behind some of the insignificant results.

In the present study, we attempted to identify the main factors that affect the improvement of tinnitus within patients with VSB implants. We found that reduction of MCL, which increases the masking effect, is an important factor in the reduction of tinnitus, and that improvement of MCL could explain 60.0% of THI improvement. The remaining 47.6% could be explained by non-audiological factors. The pathophysiology of tinnitus includes audiological factors, as well as central mechanisms and psychological factors. A previous meta-analysis reported that combined management of audiological and non-audiological factors, rather than sole masking, is more effective in reducing tinnitus [28]. This emphasizes the heterogeneity of tinnitus, and the importance of non-audiological factors in its treatment. Although tinnitus retraining therapy or direct counseling for tinnitus were not performed in the present study, a frequent fitting schedule with counseling and stress reduction as a consequence of hearing gain would qualify as combined management.

The limitations of this study included those inherent to retrospective designs and analyses. Thus, a prospective study comparing the extent of tinnitus suppression and changes in MCL between VSB and other HAs is necessary to validate the present findings. Furthermore, the sample size was small and may have resulted in type II errors, as discussed earlier. Future studies should also include larger samples to overcome this limitation. A 1-year follow-up period was sufficient to prove the beneficial effects of VSB on tinnitus; however, a longer-term follow-up period may be needed. If hearing restoration is not achieved with the VSB because of long-term aggravation of hearing loss, tinnitus reduction accompanied by sufficient hearing gain would not be achieved. Additional acoustic amplification via conventional HA or replacement with a cochlear implant could be helpful in overcoming possible long-term aggravation of hearing loss and tinnitus with VSB implants. Finally, evaluation of secondary plastic reorganization in the central auditory system might be helpful in elucidating the therapeutic effects of VSB on tinnitus.

## Conclusion

VSB showed beneficial effects on tinnitus, and its efficacy was comparable to that previously reported for other hearing devices. The changes in tinnitus were best correlated to the changes

in MCL, rather than with the amount of FHG or improvement in WRS, suggesting the importance of masking effects on tinnitus suppression.

## Supporting information

**S1 Dataset. Detailed information and results for the included patients (n = 16).**
(XLSX)

## Author Contributions

**Conceptualization:** Jeon Mi Lee.

**Data curation:** Jeon Mi Lee, Hyun Jin Lee.

**Formal analysis:** Jeon Mi Lee, Hyun Jin Lee.

**Funding acquisition:** In Seok Moon.

**Methodology:** Hyun Jin Lee.

**Supervision:** In Seok Moon, Jae Young Choi.

**Writing – original draft:** Jeon Mi Lee.

**Writing – review & editing:** In Seok Moon, Jae Young Choi.

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
