## [Decision Letter · Decision Letter 0]

20 Nov 2019

PONE-D-19-23345

Effects of Vibrant Soundbridge on tinnitus accompanied with sensorineural hearing loss

PLOS ONE

Dear Professor Moon,

Thank you for submitting your manuscript to PLOS ONE. After careful consideration, we feel that it has merit but does not fully meet PLOS ONE’s publication criteria as it currently stands. Therefore, we invite you to submit a revised version of the manuscript that addresses the points raised during the review process.

We would appreciate receiving your revised manuscript by Jan 04 2020 11:59PM. To enhance the reproducibility of your results, we recommend that if applicable you deposit your laboratory protocols in protocols.io, where a protocol can be assigned its own identifier (DOI) such that it can be cited independently in the future. For instructions see: http://journals.plos.org/plosone/s/submission-guidelines#loc-laboratory-protocols

We look forward to receiving your revised manuscript.

Kind regards,

Vinaya Manchaiah, AuD, MBA, PhD

Academic Editor

PLOS ONE

Journal Requirements:

Additional Editor Comments:

I have reviewed the manuscript in addition to an independent reviewer with expertise in this area and also a statistical advisor. We think the manuscript require some revision before considering for publication.

Reviewers' comments:

Reviewer's Responses to Questions

**Comments to the Author**

1. Is the manuscript technically sound, and do the data support the conclusions?

Reviewer #1: No

Reviewer #2: Yes

2. Has the statistical analysis been performed appropriately and rigorously? 

Reviewer #1: No

Reviewer #2: Yes

3. Have the authors made all data underlying the findings in their manuscript fully available?

Reviewer #1: No

Reviewer #2: Yes

4. Is the manuscript presented in an intelligible fashion and written in standard English?

Reviewer #1: Yes

Reviewer #2: Yes

5. Review Comments to the Author

Reviewer #1: 1. Page 2, lines 24-29: Instead of showing p-values, I suggest to depict point estimates and their 95% confidence intervals.

2. Page 5, lines 91-93: I was curious how many subjects were not included because of the lack of follow-up, audiological tests, and/or self-assessment questionnaires. Inclusion criteria #1-#3 and exclusion criteria #1-#3 would provide the number of study subjects within a 7-month period.

3. Table 2:

(3a) Column "A-B": I assumed the test for A-B comparison was a paired t test. For paired t tests, I would suggest to replace the percent change with a standard deviation of the difference. The percent change can be easily calculated by readers if necessary.

(3b) Column "Probability": I suggest to replace "probability" with 'p-value'.

4. Page 10, lines 189-198: With the sample size of 16, the regression analyses were likely to be under power, meaning that the only significant finding might not be reproducible (i.e., high type 2 error) or falsely observed (i.e., high type 1 error). Please discuss this as a limitation in the Discussion section.

5. Table 3: I suggest to replace the standard errors with 95% confidence intervals for the regression estimates.

Reviewer #2: This is a well-written manuscript that should be of interest to some of the journal's readers. I have just a few minor comments:

1. Page 4, Lines 79-80: "Thus, we hypothesized that VSB . . . " This sentence needs to be re-worded.

2. Table 2: add units for PTA and MCL (e.g., dB HL)

3. Page 9, Line 170: "The questionnaire results demonstrated statistically significant improvements . . . " Add the word "statistically"

4. Page 9, Lines 173-174: "Based on the 20% criterion, 7 patients . . . " add "20% criterion"

5. Page 10, Line 194: "only increased MCL was significantly . . . " I think the authors mean "decreased MCL"

6. Page 11, Lines 213-214: "There are no known pathophysiological accounts pertaining to . . ." This sentence should be re-worded because I'm not sure what "pathophysiological accounts" means.

7. Page 12, Lines 249-251: The last 2 sentences should be deleted because the point of this VAS was NOT to compare current conditions with the patient's memory from 1 year ago. The goal of this VAS was to assess the patient's condition at the time they answered the question: once pre-surgery, and once post-surgery.

8. Page 13, Line 261: "relief from hearing gain" Do the authors mean "relief from hearing loss"?

6. PLOS authors have the option to publish the peer review history of their article (what does this mean?). If published, this will include your full peer review and any attached files.

Reviewer #1: No

Reviewer #2: No

---

## [Author Response · Author response to Decision Letter 0]

18 Dec 2019

Reviewer #1: 

1. Page 2, lines 24-29: Instead of showing p-values, I suggest to depict point estimates and their 95% confidence intervals.

- Thank you for your valuable suggestion. We have reanalyzed the values and modified the paragraph as recommended. 

- (lines 24-31) “VAS scores for loudness (mean difference: 1.9, 95% CI: 0.6, 3.1), awareness (mean difference: 1.6, 95% CI: 0.4, 2.8), and annoyance (mean difference: 1.7, 95% CI: 0.7, 2.8) showed significant improvements from baseline to 12 months after surgery. In addition, THI scores showed a significant decrease (mean difference: 13.8, 95% CI: 2.9, 24.9). The average hearing threshold level, WRS, and most comfortable level (MCL) also showed significant improvements at 12 months after surgery (mean difference: 17.3, 95% CI: 13.3, 21.3; mean difference: −7.6, 95% CI: −15.1, −0.1; mean difference: 26.3, 95% CI: 22.9, 29.6, respectively). Among the aforementioned factors, changes in MCL were best correlated with those in THI scores (mean difference: 2.55, 95% CI: 0.90, 4.21).”

2. Page 5, lines 91-93: I was curious how many subjects were not included because of the lack of follow-up, audiological tests, and/or self-assessment questionnaires. Inclusion criteria #1-#3 and exclusion criteria #1-#3 would provide the number of study subjects within a 7-month period.

- Thank you for your question. We retrospectively reviewed data for patients who received VSB between January 2016 and July 2019. At the first medical examination, we routinely asked the patients about the presence and duration of tinnitus. Although we did not observe the patients for 2 years, the durations (>2 years) were self-reported by the patients. We agree than the current information can cause confusion and have made the relevant changes in the revised manuscript. 

- (lines 92-93) “…3) stable chronic tinnitus that did not respond to any previous treatments,…”

- (lines 98-101) “In total, 47 patients underwent VSB implantation between January 2016 and July 2019. From these, 3 patients were excluded because they had mixed hearing loss. In addition, 16 patients who did not experience tinnitus, 4 who were followed up for less than 1 year, 7 who did not complete the questionnaires, and 1 foreigner for whom a speech discrimination test was not available were excluded.” 

3. Table 2:

(3a) Column "A-B": I assumed the test for A-B comparison was a paired t test. For paired t tests, I would suggest to replace the percent change with a standard deviation of the difference. The percent change can be easily calculated by readers if necessary.

(3b) Column "Probability": I suggest to replace "probability" with 'p-value'.

- Thank you for the constructive advice. We have amended the table as per your recommendations. 

4. Page 10, lines 189-198: With the sample size of 16, the regression analyses were likely to be under power, meaning that the only significant finding might not be reproducible (i.e., high type 2 error) or falsely observed (i.e., high type 1 error). Please discuss this as a limitation in the Discussion section.

- Thank you for your valuable comments. We agree that the sample size was too small to perform regression analysis and may have resulted in type II errors. We have mentioned this in the Discussion section and added it as a limitation. 

- (lines 255-258) “Finally, a sample size compromises the power for the detection of treatment effects and increases the possibility of false-negative results. Thus, the small sample size in this study may be a reason behind some of the insignificant results.” 

- (lines 272-274) “Furthermore, the sample size was small and may have resulted in type II errors, as discussed earlier. Future studies should also include larger samples to overcome this limitation.”

5. Table 3: I suggest to replace the standard errors with 95% confidence intervals for the regression estimates.

- Thank you for the constructive advice. We have reanalyzed the values and replaced the standard errors with 95% confidence intervals in Table 3. 

Reviewer #2: This is a well-written manuscript that should be of interest to some of the journal's readers. I have just a few minor comments: 

1. Page 4, Lines 79-80: "Thus, we hypothesized that VSB . . . " This sentence needs to be re-worded.

- Thank you for the positive feedback and comments. We have rephrased the sentence in the revised manuscript. 

- (lines 81-82) “Accordingly, we hypothesized that the use of VSB as a hearing device would have a beneficial effect on tinnitus in patients with sensorineural hearing loss.”

2. Table 2: add units for PTA and MCL (e.g., dB HL)

- Thank you for pointing this out. We have added units for PTA, WRS, and MCL in Table 2. 

3. Page 9, Line 170: "The questionnaire results demonstrated statistically significant improvements . . ." Add the word "statistically"

4. Page 9, Lines 173-174: "Based on the 20% criterion, 7 patients . . . " add "20% criterion"

- Thank you for the valuable suggestions. We have modified the sentences as recommended. 

- (lines 173-174) “The questionnaire results demonstrated statistically significant improvements after VSB implantation.”

- (lines 176-178) “Based on the 20% criterion, 7 patients (43.8%) reported improvement in tinnitus loudness, 8 patients (50.0%) reported no change, and 1 patient (6.2%) reported louder tinnitus after implantation.” 

5. Page 10, Line 194: "only increased MCL was significantly . . ." I think the authors mean "decreased MCL"

- We apologize for the overlook. We have rectified the error in the revised manuscript. 

- (lines 196-198) “When we performed multiple regression analysis, only decreased MCL was significantly and positively related to the improvement of THI.”

6. Page 11, Lines 213-214: "There are no known pathophysiological accounts pertaining to . . ." This sentence should be re-worded because I'm not sure what "pathophysiological accounts" means.

- We apologize for the lack of clarity. We have rephrased the sentence and merged it with the subsequent sentence in the revised manuscript. 

- (lines 219-221) “The mechanism underlying the beneficial effect of VSB on tinnitus remains unclear, although it may be similar to that of other hearing devices.” 

7. Page 12, Lines 249-251: The last 2 sentences should be deleted because the point of this VAS was NOT to compare current conditions with the patient's memory from 1 year ago. The goal of this VAS was to assess the patient's condition at the time they answered the question: once pre-surgery, and once post-surgery.

- Thank you for the suggestion. We have deleted the sentences from the revised manuscript. 

8. Page 13, Line 261: "relief from hearing gain" Do the authors mean "relief from hearing loss"?

- We apologize for the lack of clarity. Because the pathophysiology of tinnitus includes psychological aspects, reducing emotional stress is crucial for managing the condition tinnitus. Emotional stress caused by hearing loss also affects the severity of tinnitus, and hearing gain by means of hearing devices can help in overcoming this emotional stress. Accordingly, we used the expression “relief from hearing gain”, although we agree that the phrase is not clear. We have now modified the sentence in the revised manuscript. 

- (lines 264-266) “Although tinnitus retraining therapy or direct counseling for tinnitus were not performed in the present study, a frequent fitting schedule with counseling and stress reduction as a consequence of hearing gain would qualify as combined management.”

---

## [Decision Letter · Decision Letter 1]

17 Jan 2020

Effects of Vibrant Soundbridge on tinnitus accompanied by sensorineural hearing loss

PONE-D-19-23345R1

Dear Dr. Moon,

We are pleased to inform you that your manuscript has been judged scientifically suitable for publication and will be formally accepted for publication once it complies with all outstanding technical requirements.

Also, please note that one of the reviewer suggest a very minor revision which can be done during the production stage. 

With kind regards,

Vinaya Manchaiah, AuD, MBA, PhD

Academic Editor

PLOS ONE

Additional Editor Comments (optional):

Reviewers' comments:

Reviewer's Responses to Questions

**Comments to the Author**

1. If the authors have adequately addressed your comments raised in a previous round of review and you feel that this manuscript is now acceptable for publication, you may indicate that here to bypass the “Comments to the Author” section, enter your conflict of interest statement in the “Confidential to Editor” section, and submit your "Accept" recommendation.

Reviewer #1: All comments have been addressed

Reviewer #2: All comments have been addressed

2. Is the manuscript technically sound, and do the data support the conclusions?

Reviewer #1: Yes

Reviewer #2: Yes

3. Has the statistical analysis been performed appropriately and rigorously? 

Reviewer #1: Yes

Reviewer #2: Yes

4. Have the authors made all data underlying the findings in their manuscript fully available?

Reviewer #1: No

Reviewer #2: Yes

5. Is the manuscript presented in an intelligible fashion and written in standard English?

Reviewer #1: Yes

Reviewer #2: Yes

6. Review Comments to the Author

Reviewer #1: I appreciate the authors' careful and detailed responses. The responses and the revised manuscript help me better understand the goal of the manuscript.

I have one general comment. In lines 81-82, I suggest to replace ``beneficial effect'' with `positive association'. Because this was an observational study, I suggest to avoid causal terminologies throughout the manuscript.

Reviewer #2: (No Response)

7. PLOS authors have the option to publish the peer review history of their article (what does this mean?). If published, this will include your full peer review and any attached files.

Reviewer #1: No

Reviewer #2: No

---

## [Editor Report · Acceptance letter]

22 Jan 2020

PONE-D-19-23345R1 

Effects of Vibrant Soundbridge on tinnitus accompanied by sensorineural hearing loss 

Dear Dr. Moon:

I am pleased to inform you that your manuscript has been deemed suitable for publication in PLOS ONE. Congratulations! Your manuscript is now with our production department. 

With kind regards,

on behalf of

Dr. Vinaya Manchaiah 

Academic Editor

PLOS ONE